# Identification of potential biomarkers associated with immune cell infiltration patterns in Kawasaki disease via bioinformatics

**Guolian Wu[1], Hui Liu[2], Meiling Wang[1], Rong Wang** 🟢[1]*

**1** Department of Pediatrics, Cangzhou People's Hospital, Cangzhou, Hebei Province, China,
**2** Department of Hematology, Cangzhou Central Hospital, Cangzhou, Hebei Province, China

* 19933278898@163.com

## Abstract

Inflammation and immune dysregulation play critical roles in Kawasaki disease (KD) pathogenesis, yet specific biomarkers and immune signatures remain elusive. This study aims to identify key biomarkers and characterize immune cell infiltration scores in KD using bioinformatic approaches. The GSE73461 dataset, downloaded from the Gene Expression Omnibus (GEO) database, includes 78 KD patients and 55 normal controls collected by Imperial College London from 2015 to 2023, and was analyzed to identify differentially expressed genes (DEGs). Gene ontology (GO) and Kyoto Encyclopedia of Genes and Genomes (KEGG) analysis revealed significant involvement of these DEGs in acute inflammatory responses, plasma membrane components, PI3K-Akt signaling, and cytokine interactions. Protein-protein interaction (PPI) network was constructed, and five candidate hub genes (*AURKB, BUB1, CCL2, IL-4*, and *TOP2A)* were identified. Immune cell infiltration analysis gusing the XCell algorithm showed increased levels of Monocytes, neutrophils, and other immune cells in KD, while B cells and T cells were decreased. Correlation analysis indicated that these candidate hub genes are associated with immune dysregulation and inflammation in KD. These findings provide potential diagnostic biomarkers and therapeutic targets for KD, warranting further validation in larger studies.

## Introduction

Kawasaki disease (KD), first described in 1967 by Japanese pediatrician Tomisaku Kawasaki, is an acute vasculitis primarily affecting children aged 6 months to 5 years [1–3]. The most prominent complication of KD is the development of coronary artery aneurysms (CAAs), which are the most common acquired heart disease among children globally [1,4,5]. The pathogenesis of KD is closely related to various factors, including infection, genetic susceptibility, and immune response, leading to significant inter-individual disease heterogeneity and diagnostic challenges. Multiple cytokines in the innate immune system of KD patients may induce coronary inflammation in

**Data availability statement:** All relevant data are within the paper and its Supporting information files.

**Funding:** The author(s) received no specific funding for this work.

**Competing interests:** The authors have declared that no competing interests exist.

response to pathogen invasion [5]. Meanwhile, the adaptive immune response is significantly activated. Recent studies have shown that pro-inflammatory and regulatory T cells in the blood play critical roles in regulating the severity and susceptibility to KD [5]. However, KD patients often lack apparent clinical manifestations, leading to delayed diagnosis and treatment [6].

Despite significant advancements in molecular biomedicine and extensive research, the primary etiology of KD remains elusive [4,6]. Current understanding of KD pathophysiology suggests that it involves the initiation of inflammatory cascade reactions and the activation of both the innate and adaptive immune systems [3,7]. Numerous studies have identified key cytokines associated with KD, including IL-1, IL-18, IL-6, TNF-α, IFN-γ, TNF-β, IL-10, and IL-8, have been confirmed to be associated with KD [1,3]. During the acute phase of KD, the innate immune system appears to play a dominant role, with clear evidence of adaptive immune activation. Peripheral blood analysis reveals a marked increase in neutrophils, monocytes, and macrophages, while lymphocyte proportions and numbers decrease [4]. Previous study on adaptive immune cells focuses on the features of B cells, CD4 + T cell, CD8 + T cells, Tregs and Th17 cells, yet their specific changes and functions in KD remain poorly understood 4. Given these gaps in knowledge, there is an urgent need to delve deeper into the pathophysiological mechanisms of KD using innovative approaches. Bioinformatics analysis, which can effectively analyze biomarkers and cellular changes, has shown great potential in various fields and may provide valuable insights into KD.

In this study, microarray datasets related to KD were retrieved from the Gene Expression Omnibus (GEO) database [4]. Through comprehensive bioinformatic analysis, we identified potential biological markers and cell infiltration patterns in KD specimens compared to normal controls (NCs). Afterwards, principal component analysis (PCA) was employed to reduce the dimensionality of infiltrating cells while preserving critical information [6,8]. This study aims to identify corresponding biomarkers and immune cell infiltration in KD, thereby providing insight for early diagnosis and targeted therapeutic strategies [6,8].

## Method

### Data access and Screening of Differentially Expressed Genes (DEGs)

GSE73461 was acquired from the Gene Expression Omnibus database (GEO) (http://www.ncbi.nlm.nih.gov/geo/), and constructed based on GPL10558 Illumina HumanHT-12 V4.0 expression beadchip platform and made up of 78 KD and 55 normal control (NC) individuals [9]. Data preprocessing was performed using the limma package in R (version 4.3.0) [10].

The purpose of screening DEGs is to identify genes with significant expression changes under different conditions. A gene expression matrix was transformed from the probe expression through the "org.Hs.e.g.,db" R package using a platform annotation file [5]. Limma packages in R program were employed to screen the DEGs between KD and NC ($|$ log2 fold change (FC) $| \geq 2$ and an adjusted false discovery rate (FDR) $P < 0.05$).

## Functional and pathway enrichment analysis with redundancy handling

To achieve functional and pathway interpretation of gene clusters, we used the "clusterProfiler" R package for GO and KEGG enrichment analyses to identify significant functions and pathways related to DEGs [11]. To minimize redundancy, we set a redundancy threshold of 0.7, merging pathways with >70% gene overlap. We also used $P < 0.05$ and $q < 0.4$ to ensure significance. Results were visualized with the "ggplot2" package [5].

## Construction of PPI network and filtration of Hub genes

To identify interactions among DEGs, a Homo sapiens PPI network was built with a confidence threshold ≥0.7, hiding unconnected nodes (http://string-db.org). The purpose of filtration of candidate hub genes is to identify key genes with significant interactions in biological networks. Candidate hub genes were identified as those with degree centrality above the 90th percentile from the PPI network constructed using the igraph R package.

## Data normalization and immune cell infiltration analysis

To prepare the GSE73461 dataset, we removed genes with zero expression and applied scaling normalization. This process involved removing IL4 due to consistently low expression post-normalization, ensuring data quality for subsequent cell infiltration analysis. We then used the XCell algorithm to evaluate infiltration scores for 29 immune and non-immune cells [12], identifying significant differences between subgroups at $P < 0.05$. Finally, we visualized the results using heatmaps and violin plots generated by the R packages "pheatmap" and "vioplot" to compare the KD and NC groups [13].

## Correlation analysis and PCA

To evaluate the correlation between candidate hub genes and infiltrating immune cells, and to assess the efficacy of a node-gene signature in distinguishing KDs from NCs, we used the Spearman correlation method ($P < 0.05$). Correlation coefficients were categorized as follows: strong ($|r| \geq 0.7$), moderate ($0.4 \leq |r| < 0.7$), and weak ($0.1 \leq |r| < 0.4$). The correlation heatmap was visualized using the "ggplot2" package. To enhance the robustness of the PCA analysis, we identified and removed an outlier that deviated significantly from the main data cluster (PCA1 > 4 and PCA2 < -15). The PCA plot, generated with "limma" and "ggplot2", now accurately depicts KDs and NCs separation based on the node-gene signature [3,7,14].

# Results

## Identification of DEGs

In this study, we identified 1229 DEGs with with $P < 0.05$ and $|\log2 \text{FC}| \geq 2$, including 720 up-regulated and 509 down-regulated genes. Notably, those with $|\log2 \text{FC}| > 8$ are particularly important for understanding specific biological processes or disease states and require further study (Fig 1).

## GO and KEGG enrichment analysis

GO and KEGG analysis highlight key biological roles of DEGs. GO analysis shows significant links to acute inflammatory response and plasma membrane components (Fig 2A and S1A Fig), which are expected given KD's characteristic inflammation and vascular involvement. KEGG analysis points to enrichment in pathways like cytokine interactions and PI3K-Akt signaling (Fig 2B and S1B Fig), further supporting KD's known inflammatory and immune-mediated nature. While the detailed enrichment patterns offer a more comprehensive view of its molecular mechanisms. Fig 2C, D show how DEGs relate to these pathways, emphasizing their importance in our study's biological context and offering new insights into potential therapeutic targets.

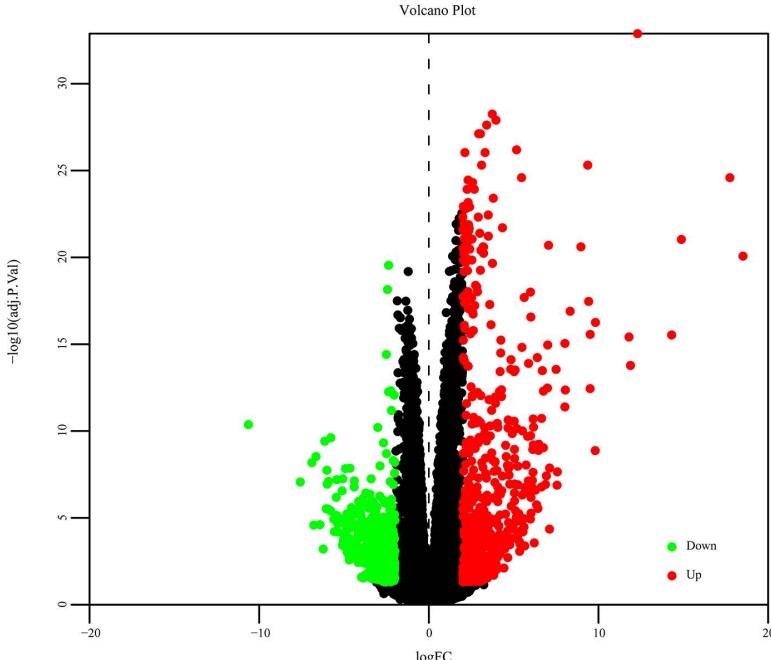

**Fig 1. Volcano plot depicting differentially expressed genes (DEGs) comparison between KDs and NCs.**

## PPI network construction and hub gene identification

To pinpoint critical genes contributing to KD, we constructed a PPI network. Nodes of 876 intersecting genes were detected in the PPI network using a medium confidence level of ≥0.7 (Fig 2E). Our analysis using an R package reveals five prominent candidate hub genes: AURKB, BUB1, CCL2, IL4, and TOP2A (Fig 2F). These genes, central to the network's structure, exhibit significant expression levels, underscoring their pivotal role in KD pathogenesis.

## Immune cell infiltration analysis

Employing the XCell algorithm, we quantified 29 distinct cell types across samples. S2 Fig A shows the relative abundance of cell composition in two groups. S2 Fig B shows that in KD, CD4 + T cells and CD8 + T cells, B cells, and Th1 cells are decreased, while Megakaryocytes, platelets, basophils, aDCs, NKT cells, Monocytes, and neutrophils are increased compared to NC. S2 Fig C shows that CD8 + T cells are positively associated with CD4 + T cells, CD8 + Tcm cells, and CD8 + Tem cells, and CD8 + Tcm cells correlates with CD8 + Tem cells, while both are negatively correlated with Monocytes and neutrophils. B-cells are positively associated with Memory and Naive B-cells. Strong positive relationships are also observed between HSC and iDCs, Megakaryocytes and platelets, and Monocytes and neutrophils, suggesting cell interactions. PCA in S2 Fig D distinguishes KD from NC, indicating immune dysregulation in KD pathogenesis.

## Correlation analysis

Fig 3A shows KD patients have higher levels of aDC, basophils, eosinophils, HSC, Megakaryocytes, Monocytes, neutrophils, platelets, and NKT cells, but lower levels of B cells, CD4+ and CD8 + T cells, cDC, Memory B cells, CLP, MSC, Naive B cells, Plasma cells, and Th1 cells. Fig 3B reveals significant correlations: AURKB, BUB1, and TOP2A are linked to Th1 cells, Plasma cells, and Smooth muscle, highlighting their immune roles, while showing negative ties to Naive

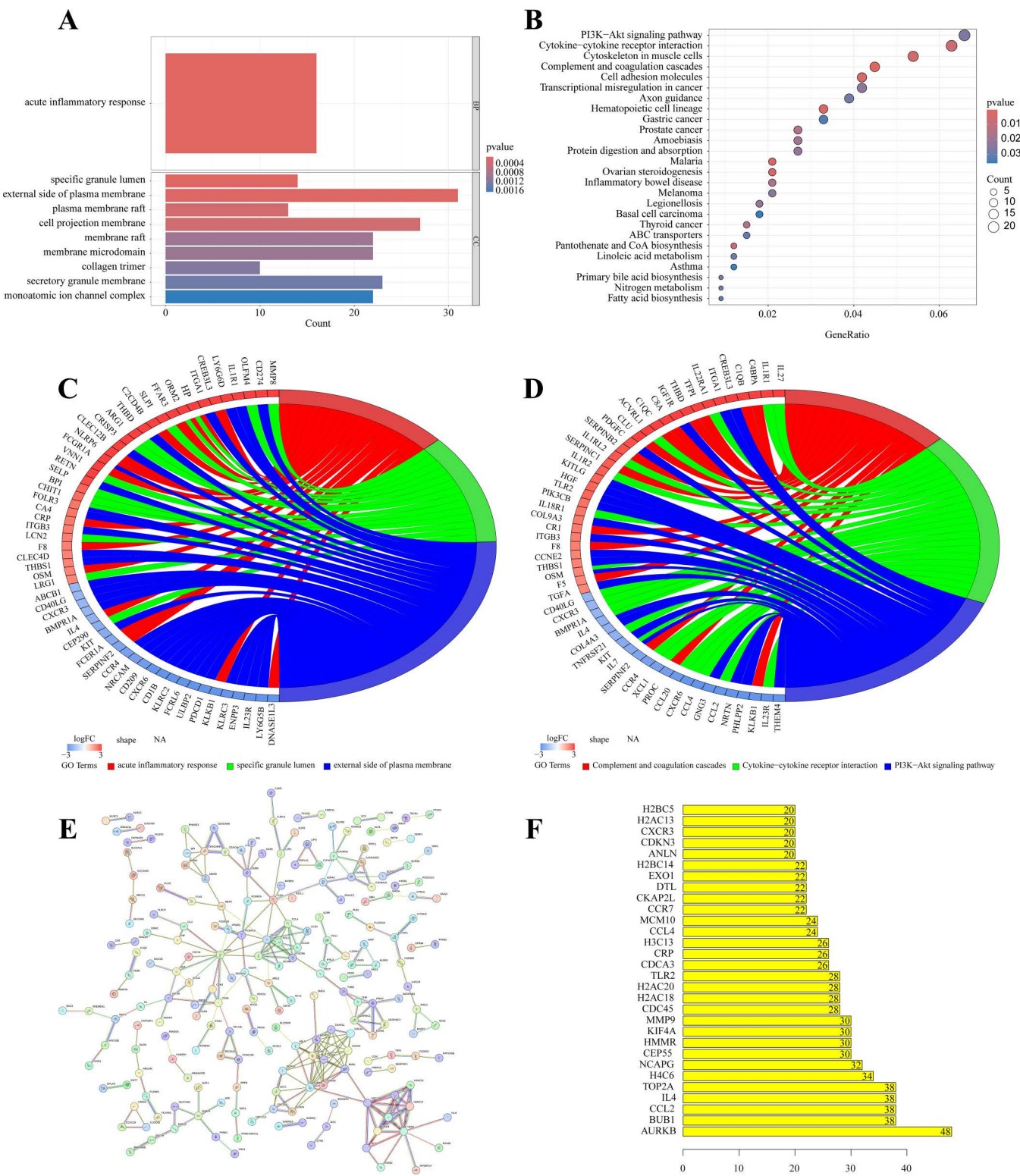

**Fig 2. (A) Gene Ontology (GO) analysis of differentially expressed genes (DEGs).** (B) Kyoto Encyclopedia of Genes and Genomes (KEGG) pathway analysis of DEGs. (C, D) DEGs are enriched in the top three GO and KEGG terms, with colors representing distinct functions and pathways. (E) Visualization of the Protein-Protein Interaction (PPI) network among DEGs. (F) Identification of candidate hub genes to highlight key genes' significance.

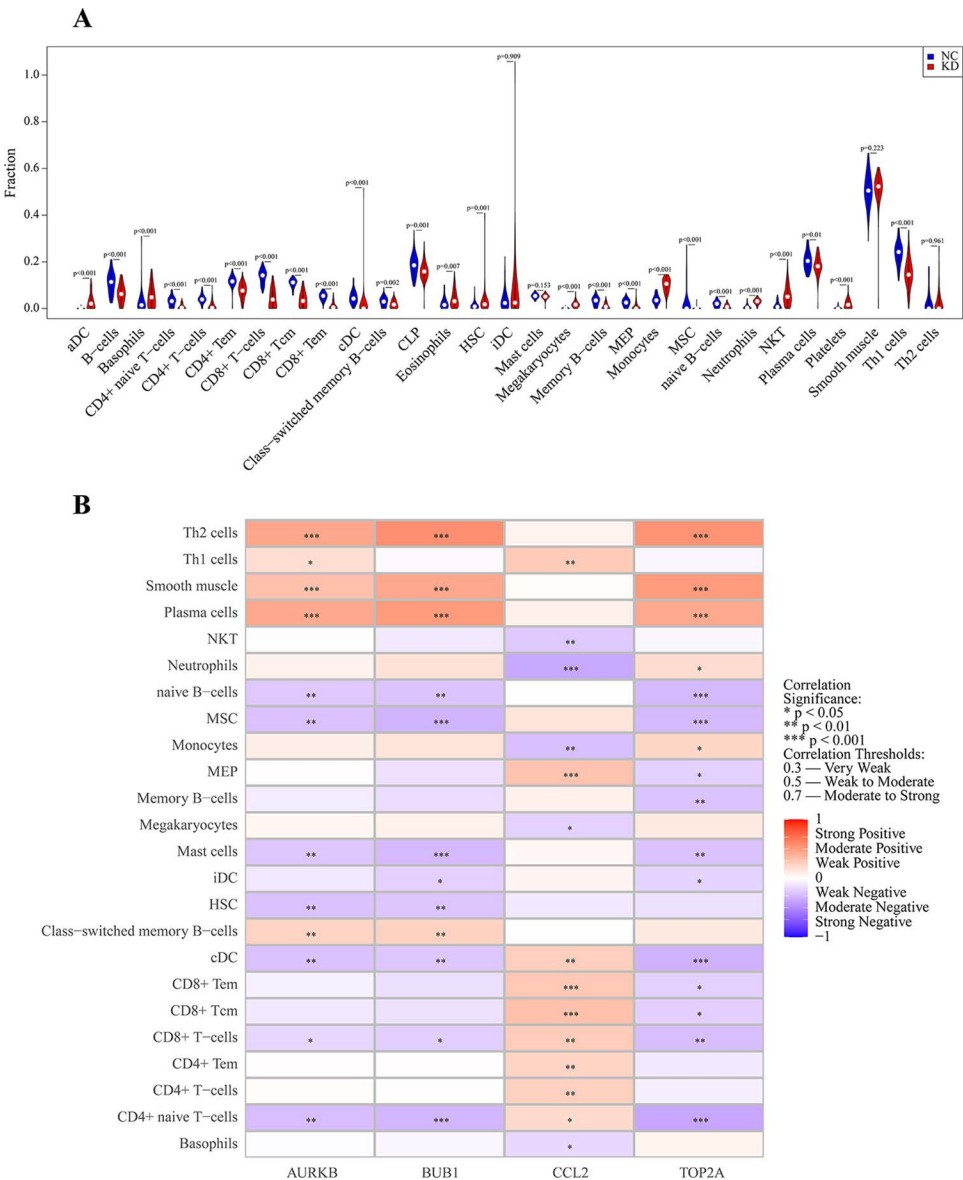

**Fig 3. (A) This violin plot illustrates the differences in immune and non-immune cell infiltration levels between KDs and NCs.** (B) The heatmap visualizes the correlation between key genes and immune and non-immune cell types.

B-cells, MSCs, mast cells, cDCs, CD8 + T cells, and CD4 + Naive T cells. CCL2 is positively associated with T-cell subsets but negatively with NKT cells, neutrophils, Monocytes, Megakaryocytes, and basophils, indicating complex immune interactions in KD.

## Discussion

KD is a major cause of acquired heart disease in children, primarily characterized by acute vasculitis targeting the coronary arteries. Although the precise cause of the disease is still unknown, it is believed to be initiated by an unidentified stimulus that sets off an immune-mediated inflammatory response in children with genetic predispositions [3]. Therefore,

investigating gene expression and the roles of key signaling pathways and cellular profiles is crucial for understanding KD pathogenesis. In this study, an integrated bioinformatics analysis identified four candidate hub genes that show potential as diagnostic biomarkers for KD.

Our study identified 1,229 DEGs in KD patients, with 720 up-regulated and 509 down-regulated. GO analysis showed that DEGs are enriched in acute inflammatory responses and plasma membrane components, indicating potential involvement in inflammatory cell infiltration, membrane remodeling, and signal transduction. In KD, the inflammatory response is characterized by fever, rash, lymphadenopathy, and coronary artery inflammation [3]. In the acute phase of KD, Monocytes and neutrophils infiltrate and release cytokines such as IL, IFN-γ, and TNF-α, amplifying inflammation [8,15]. Platelet-monocyte aggregation further promotes pro-inflammatory responses, exacerbating the inflammatory cascade and worsening coronary artery damage [16]. Plasma membrane components—lipids, proteins, carbohydrates, and extracellular matrix (ECM)-interacting molecules—form a complex structure crucial for signal transduction, cell-cell interactions, and cellular functions. Study shows Ubiquitin-specific protease 5 (USP5) regulates inflammatory cytokines, indirectly modulates plasma membrane receptor function, enhances inflammatory cell adhesion and migration, and promotes ECM remodeling, thereby exacerbating coronary artery damage [17]. These results are consistent with the vasculitis and immune activation observed in KD [18,19]. KEGG analysis further revealed DEGs enrichment in the PI3K-Akt pathway and cytokine-cytokine receptor interaction (CCRI), consistent with previous results [20,21], which are critical for cell growth and immune regulation. The PI3K/AKT signaling pathway is key in cell migration, epithelial-mesenchymal transition (EMT), and protecting endothelial cells from inflammation, making it highly relevant in KD, where vascular injury and coronary artery lesions (CAL) are primary concerns [20,22,23]. Cytokines are signaling molecules produced by immune and non-immune cells, encompassing interleukins, tumor necrosis factors, colony-stimulating factors, interferons, chemokines, and growth factors. They bind to specific receptors, triggering cell apoptosis, development, differentiation, immune regulation, inflammation, and tissue repair [24]. The CCRI pathway may influence the progression of coronary arteritis in KD by regulating the infiltration of inflammatory cells and the release of cytokines [25]. These results suggest DEGs may play a role in KD's inflammation and immune activation via PI3K-Akt and CCRI signaling pathways. So targeting this pathway could be a novel therapeutic strategy for preventing and managing vascular injury and CAL in KD patients.

In our study, we identified four candidate hub genes—AURKB, BUB1, TOP2A, and CCL2—that may play crucial roles in the pathology of KD. AURKB, BUB1, and TOP2A are all involved in cell cycle regulation and chromosomal stability. Specifically, AURKB and BUB1 are mitotic regulators essential for cell division and proper chromosome segregation [26,27], while TOP2A is a DNA topoisomerase crucial for DNA replication and repair [28]. Although the three genes are not enriched in the PI3K/AKT signaling pathway, their high expression impacts this pathway [22,29,30], which alleviates coronary artery disease via macrophage autophagy [23]. Meanwhile, dysregulation of the PI3K/AKT signaling pathway and aberrant expression of these genes can lead to DNA damage, cellular stress, and chromosomal instability, which may contribute to endothelial cell proliferation and damage observed in KD [31]. CCL2 exhibits dual roles in inflammation and tissue repair. It recruits inflammatory cells (e.g., Monocytes and neutrophils) via the CCRI pathway through its receptor CCR2, and activates the Dectin-2/NLRP3 inflammasome to induce IL-1β, driving vasculitis in KD models [32]. Conversely, it promotes cardiomyocyte proliferation via the JNK/STAT3 pathway, aiding cardiac regeneration and repair after myocardial infarction [33]. In summary, the candidate hub genes identified (AURKB, BUB1, TOP2A, and CCL2) impact KD pathology via cell cycle regulation, inflammation, and tissue repair, with potential impacts on PI3K/AKT signaling and CCRI pathway, suggesting their potential as therapeutic targets.

The results show that in KD samples, the levels of aDCs, basophils, eosinophils, HSCs, Megakaryocytes, Monocytes, neutrophils, platelets, and NKT cells are significantly higher, which is consistent with previous research [19]. In KD, neutrophils initially infiltrate inflamed sites, releasing pro-inflammatory factors that exacerbate endothelial cell damage [34]. Subsequently, Monocytes and macrophages are recruited to the inflamed areas by chemokines, where they secrete cytokines (such as TNF-α, IL-1β, and IL-6) that amplify the inflammatory response [35]. Dendritic cells can activate T cells to

initiate adaptive immune responses [36]. Platelet increase is a key sign of KD, important for diagnosis and monitoring [37]. Single-cell sequencing found more NKT cells in KD patients, suggesting they might play a role in KD's immune response [38]. Eosinophils and basophils may be related to KD pathology, and their counts could be key factors in a machine learning model to predict KD from routine blood tests [39]. HSCs' role in KD is linked to disease progression through immune and inflammatory effects [40]. The interactions among these immune cells form the core mechanism of the inflammatory response in KD. It's important to note that our analysis showed changes in basophils, eosinophils, and HSCs during KD. Since there are currently less studies on these cells in relation to KD, our findings could potentially offer new directions for future research into the role of these cells in KD.

In this study, candidate hub genes AURKB, BUB1, and TOP2A significantly correlate with multiple immune and non-immune cells. This correlation analysis was performed to explore the interactions between immune cells and candidate hub gene expression, offering insights into their roles in KD. AURKB and BUB1's abnormal expression can disrupt cell cycle regulation, influencing the proliferation and functions of endothelial and immune cells to regulate angiogenesis [41,42], potentially affecting vascular inflammation in KD. High TOP2A expression may impair DNA repair, increasing cellular stress and inflammation, possibly activating inflammatory pathways like PI3K/AKT, linked to KD development [22]. CCL2's negative correlation with neutrophils, Monocytes, and Megakaryocytes, and positive correlation with T cells, indicates its complex role in KD. It may recruit cells during inflammation and promote T cell activation and cardiomyocyte proliferation, potentially promoting coronary artery repair in KD's recovery phase [32,33]. All of these shows AURKB, BUB1, TOP2A, and CCL2 play key roles in KD through affecting cell functions and inflammation response, and may aid in coronary artery repair, making them important for treatment.

## Conclusion

This study identified four key genes—AURKB, BUB1, TOP2A, and CCL2—that are involved in KD and could serve as diagnostic biomarkers. We also discovered unique immune cell patterns that offer new insights into KD's immunopathology. These genes, interacting with immune cells and linked to the PI3K/AKT and CCRI pathways, present new targets for KD treatment. While these findings are promising, they require further validation in larger studies and deeper investigation to develop effective clinical applications for KD.

This study has several limitations. The small sample size and single KD dataset may not fully capture KD patient diversity, affecting result generalizability. The study also doesn't accurately show gene expression patterns and immune cell infiltration in different KD subtypes, limiting our understanding of KD pathogenesis. Additionally, the study lacks experimental validation and has not evaluated the specificity and sensitivity of the identified potential biomarkers for KD. Future studies should use larger sample size, more diverse datasets, validate findings in independent cohorts, and conduct experiments to confirm the biological relevance of identified candidate hub genes.

## Supporting information

**S1 Fig.  Figure 1.**
(TIF)

**S2 Fig.  Figure 2.**
(TIF)

**S1 File.  Download Data.**
(RAR)

**S2 File.  Organize Data.**
(ZIP)

**S3 File. Grouping and Sorting.**
(ZIP)

**S4 File. Differential Analysis.**
(ZIP)

**S5 File. Gene Name to ID Conversion.**
(ZIP)

**S6 File. GO Enrichment Analysis.**
(ZIP)

**S7 File. GO Diagram.**
(ZIP)

**S8 File. KEGG Enrichment Analysis.**
(ZIP)

**S9 File. KEGG Diagram.**
(ZIP)

**S10 File. PPI.**
(ZIP)

**S11 File. Hub Genes.**
(ZIP)

**S12 File. Data Normalization.**
(ZIP)

**S13 File. XCell Algorithm for Immune Cell Matrix.**
(ZIP)

**S14 File. Filter Samples with p < 0.05.**
(ZIP)

**S15 File. Bar Chart.**
(ZIP)

**S16 File. Heatmap.**
(ZIP)

**S17 File. Correlation Heatmap.**
(ZIP)

**S18 File. Violin Plot.**
(ZIP)

**S19 File. PCA.**
(ZIP)

**S20 File. Immune Cell Correlation Analysis.**
(ZIP)

## Author contributions

**Conceptualization:** Guolian Wu, Hui Liu, Rong Wang.

**Data curation:** Hui Liu, Rong Wang.

**Investigation:** Meiling Wang.

**Methodology:** Rong Wang.

**Project administration:** Guolian Wu, Meiling Wang.

**Writing – original draft:** Guolian Wu, Rong Wang.

**Writing – review & editing:** Guolian Wu.

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
