## [Decision Letter · Decision Letter 0]

28 Jan 2025

PONE-D-24-55722Identification of AURKB , BUB1 , CCL2 , and TOP2A  as biomarkers, and immune infiltration in kawasaki disease through bioinformatics analysis: a preclinical studyPLOS ONE

Dear Dr. Wang,

Thank you for submitting your manuscript to PLOS ONE. After careful consideration, we feel that it has merit but does not fully meet PLOS ONE’s publication criteria as it currently stands. Therefore, we invite you to submit a revised version of the manuscript that addresses the points raised during the review process.

Please respond to reviewers' comments individually.

We look forward to receiving your revised manuscript.

Kind regards,

Xiaosheng Tan

Academic Editor

PLOS ONE

Journal Requirements:

4. In the online submission form, you indicated that [Data will be available on reasonable request.].

Reviewers' comments:

Reviewer's Responses to Questions

**Comments to the Author**

1. Is the manuscript technically sound, and do the data support the conclusions?

Reviewer #1: Yes

Reviewer #2: Yes

Reviewer #3: Partly

Reviewer #4: Partly

2. Has the statistical analysis been performed appropriately and rigorously? 

Reviewer #1: I Don't Know

Reviewer #2: Yes

Reviewer #3: Yes

Reviewer #4: Yes

3. Have the authors made all data underlying the findings in their manuscript fully available?

Reviewer #1: Yes

Reviewer #2: Yes

Reviewer #3: Yes

Reviewer #4: Yes

4. Is the manuscript presented in an intelligible fashion and written in standard English?

Reviewer #1: Yes

Reviewer #2: No

Reviewer #3: No

Reviewer #4: Yes

5. Review Comments to the Author

Reviewer #1: Thank you for your invitation.

The manuscript is about the Identification of AURKB , BUB1 , CCL2 , and TOP2A  as biomarkers, and immune infiltration in kawasaki disease through bio informatics analysis. It is of certain value, the structure is clear and the charts are appropriate, but please consider some details as outlined below:

1.Please check through the manuscript for writing. There are differences in the use of statistical notation throughout the manuscript. "p" in line 74 and 78 should be italicized, and in line 99 presented as "p-values". In line 87, "xCell" should be capitalized as "XCell", etc.

2.When using GO and KEGG to identify the enriched functions and pathways, how do you handle the issue of functional redundancy to prevent over-interpretation caused by a single gene's involvement in multiple pathways?

Reviewer #2: This manuscript presents a bioinformatics study that identifies potential biomarkers (AURKB, BUB1, CCL2, and TOP2A) and investigates immune cell infiltration patterns in Kawasaki disease (KD). The study offers valuable insights into the molecular mechanisms underlying KD and provides evidence for the potential use of these biomarkers in diagnosis and targeted therapy. However, there are several areas where the manuscript could be improved in terms of structure, clarity, language, and scientific depth. (1)The manuscript lacks a clear, cohesive structure in some sections, particularly in the Results and Discussion. There is frequent overlap, repetition, and mixing of topics;(2) The manuscript contains frequent grammatical errors, awkward phrasing, and informal expressions, which reduce its readability and professionalism. (3)Many ideas, such as the roles of AURKB and BUB1 in mitosis or the enrichment of immune-related GO terms, are repeated across sections without adding new insights. This leads to redundancy. (4) While the manuscript provides detailed descriptions of hub gene functions and pathways, the relevance to KD is not consistently emphasized. Discussions about cancer-related roles of AURKB, BUB1, and TOP2A dominate, while their specific roles in KD are underexplored. Below are partial problems listed in the order they are stated in the paper

1.Use Greek letters consistently (e.g., TNF-α instead of TNF-a) to adhere to academic conventions.

2. 'However, patients do no show apparent clinical manifestations, thus delaying diagnosis and treatment.' better changes to 'However, KD patients often lack apparent clinical manifestations, leading to delayed diagnosis and treatment.'

3. Some expressions could be made more concise and formal to enhance the academic tone of your writing.

"The data were cleaned using Limma11packages in R software (4.3.0)." , it could be better revised as "Data normalization and preprocessing were performed using the limma package in R (version 4.3.0)."

4. "log fold change" should be clarified as "log2 fold change" to be consistent with standard terminology.

5. The current structure is mostly fine, but for better readability and organization, please start each subsection with a brief explanation of the purpose of the analysis. For example:In 2.2, briefly state the purpose of constructing the PPI network (e.g., to identify interactions between differentially expressed genes), etc.

6. Some expressions are too general or lack specificity. Consider the following refinements:

2.2: PPI Network Construction:

Specify the STRING database threshold (e.g., combined score cutoff for significance);

Explain how hub genes were identified (e.g., via centrality metrics like degree or betweenness).

Include the version of Cytoscape for reproducibility.

2.3: Immune Cell Infiltration Analysis:

Shorten and simplify the description of xCell without losing its technical essence.

State how input data were normalized or preprocessed before applying xCell.

Reorganize for better readability: explain the analysis, significance threshold, and visualization in sequence.

2.4: Correlation Analysis and PCA

Specify "hub genes" and "immune cells" explicitly in the correlation analysis description.

Use consistent package names (e.g., "ggplot2" in backticks).

For Spearman analysis: indicate how correlation coefficients were interpreted (e.g., thresholds for "strong" or "weak" correlations).

7. The logical flow of the results section is unclear, and the connections between subsections are weak and the results feel fragmented, and there is no clear summary of key findings in some sections. Please reorganize the results section to create a more coherent flow. Use a logical order such as:

--Identification of DEGs: Provide the results of DEG analysis and briefly explain its significance.

--GO and KEGG Enrichment Analysis: Highlight the biological relevance of the enriched pathways and terms.

--PPI Network Construction and Hub Gene Identification: Emphasize the key findings related to hub genes.

--Immune Cell Infiltration Analysis: Explain the immune landscape differences in KD.

--Correlation Analysis and PCA: Highlight the relationships between immune cells and hub genes, and show how PCA distinguishes KD and NC samples.

8. Replace vague phrases like "immune infiltration" with "immune cell infiltration scores."

9. Section 3.3 is better suited as a methods description rather than a presentation of results.

10. In Fig. 3E, the bottom-right point in the PCA plot may be treated as an outlier and removed before further analysis.

11. The discussion lacks a clear structure, making it difficult to follow the key findings and their implications. There is no clear progression from specific results to broader implications.Multiple ideas (e.g., hub gene functions, immune mechanisms, PI3K-Akt pathway) are mixed within the same paragraph without proper separation, which makes the discussion overwhelming and repetitive. Please restructure the discussion into clearly defined subsections for better readability.

12.The discussion sometimes strays from Kawasaki disease (KD) and focuses too much on general roles of genes (e.g., their roles in tumor progression or mitosis). While these are important, the relevance to KD is not always clearly established. For example, the discussion on IL-4 mentions IVIG studies in mice but does not sufficiently connect these findings to KD in humans.

13. "It can be conclused that the four-gene above mentioned take part in the course of immune cell-mediated development in KD patients." (grammatical and awkward phrasing)

14.The conclusion is too brief and does not adequately summarize the key findings or their broader implications. Please provide a stronger, more detailed conclusion that highlights the study’s key contributions and potential impact:

--Recap the four hub genes and their potential as diagnostic biomarkers.

--Emphasize the unique immune cell infiltration patterns identified in KD.

--Discuss how the findings provide a foundation for future research into KD mechanisms and therapeutic strategies.

Reviewer #3: In this study, the authors looked at up- and downregulated genes in Kawasaki disease (KD) patients compared to healthy patients. They clustered the genes by pathways revealing immune system-related processes, as expected in KD, and PI3K-Akt signaling as potentially important for KD. Further protein-protein interaction analysis led to the identification of AURKB, BUB1, CCL2, IL4 and TOP2A as potential protein biomarkers for KD. After looking at the genes, they looked at the immune cell distribution suggesting that innate immune cells are more present in patients with KD, but adaptive immune cells are less present. They ended the study by using PCA to look at relationship between immune cell levels and gene expression levels for the previously identified ones potentially involved in KD.

While identifying biomarkers for KD is of great need due to the lack of KD-specific biomarkers at the moment, the reviewer believes that this study lacks discussion and critical analysis of the results.

Here are some major comments:

1) The rationale for correlating immune cells and gene expression, and how the link between both supports the role of the gene-associated proteins in KD remains unclear.

2) The study limitations such as the size of the dataset, the usage of a single KD dataset, the specificity/sensitivity for KD of these potential biomarkers or the lack of experimental validation needs to be discussed.

3) The language needs to be more conservative as there is no proof that the identified hub genes are valid biomarkers, while the study definitely is supporting their role in KD. For example, line 148 “In this study, five hub genes involved in KD (AURKB, BUB1, CCL2, IL-4 and TOP2A) 148 were identified.” could be “In this study, five candidate hub genes for KD (AURKB, BUB1, CCL2, IL-4 and TOP2A) were identified.”.

4) Similarly, the title needs to be changed as this is far from qualifying as a preclinical study without any experimental data to validate the findings.

5) The discussion needs to rely more on previous literature and link the role of identified hub genes with KD as the link remains unclear.

6) In general, the study is not fluid to read, with regular typos/language mistakes making it difficult to read. I would suggest taking advantage of professional language editing.

Here are some other comments:

7) Line 10: There is a missing noun word for the adjective “immune”

8) Line 12: Please clarify what “immune infiltration situation” refers to.

9) Line 14: Please add more details about GSE73461 such as specifying how many KD and control patients were included, where are they from, when was this set obtained.

10) Line 62: There are two “provide(s)”, please remove one.

11) Lines 99-101: The result part would benefit from more description of gene up- and downregulation: in the volcano plot some look significantly more upregulated and downregulated, can you comment on that? |log FC| > 8 or 10.

12) Fig 1B: The heatmap figure does not help the reader to understand the study, the results can be visualized in Fig 1A. I would suggest removing Fig 1B or moving it to the supplementary information.

13) Lines 103-108: I would suggest commenting on the results, as some of them were to be expected for KD.

14) Fig. 2: The images are too small, please ensure the text is readable.

15) Fig. 2A-2F: I would suggest moving some of the figures to a supplementary information document as some of the information is redundant why not uninteresting. This would allow for larger figures and facilitate readability of the manuscript.

16) Fig 2G: It is impossible to read it due to the size and the identified hubs are then unclear. Please update figure.

17) Fig. 3: I would suggest moving some of the figures (A, B, C, E) to the SI and maybe generating a summary figure to make the results easy to interpret for the reader supporting the statements from line 119 to line 139. Also noting that the immune cell name formatting needs to be corrected.

18) Fig. 3D: The color legend is missing, please add it to the figure.

Reviewer #4: 1) The paper lacks novelty. There are already published papers related to the same topic of the paper. For example, there is a peer reviewed published paper here https://pubmed.ncbi.nlm.nih.gov/37837870/ named Identification of hub biomarkers and immune-related pathways participating in the progression of Kawasaki disease by integrated bioinformatics analysis. The paper conducted more comprehensive and used more datasets to identify biomarkers for Kawasaki disease.

2) The methodology is too simple, just differential expression analysis, resulting in more than one thousand genes. Additionally, selecting hub genes from the protein-protein interaction (PPI) network is insufficient for establishing reliable biomarkers.

3) The paper only used a single microarray dataset, which is outdated and may not reflect current advancements in technology or data availability.

6. PLOS authors have the option to publish the peer review history of their article (what does this mean? ). If published, this will include your full peer review and any attached files.

**Do you want your identity to be public for this peer review?** For information about this choice, including consent withdrawal, please see our Privacy Policy .

Reviewer #1: No

Reviewer #2: No

Reviewer #3: No

Reviewer #4: No

---

## [Author Response · Author response to Decision Letter 1]

2 Apr 2025

Revised Submission PONE-D-24-55722: Identification of Potential Biomarkers associated with Immune Cell Infiltration Patterns in Kawasaki Disease via Bioinformatics

Dear Dr. Tan and the Editorial Team,

Thank you very much for the detailed and constructive comments from Reviewer #1, Reviewer #2, Reviewer #3, and Reviewer #4 on our manuscript. We have taken these suggestions very seriously and have made substantial revisions to address the issues raised, with particular attention to the clarity, methodology, and relevance of our findings. Here is a summary of our responses:

Reviewer #1: Thank you for your invitation.

The manuscript is about the Identification of AURKB , BUB1 , CCL2 , and TOP2A as biomarkers, and immune infiltration in kawasaki disease through bioinformatics analysis. It is of certain value, the structure is clear and the charts are appropriate, but please consider some details as outlined below:

Response:

Thank you for your positive feedback and for acknowledging the value, clarity of structure, and appropriateness of the charts in our manuscript. We have updated the title to more accurately reflect the content of our study, which is now titled "Identification of Potential Biomarkers associated with Immune Cell Infiltration Patterns in Kawasaki Disease via Bioinformatics."

We have carefully considered your comments and have made the following revisions to address your points:

Comment 1: Please check through the manuscript for writing. There are differences in the use of statistical notation throughout the manuscript. "p" in line 74 and 78 should be italicized, and in line 99 presented as "p-values". In line 87, "xCell" should be capitalized as "XCell", etc.

Response:

Thank you for your thorough review and for bringing these important points to our attention. We have carefully reviewed the manuscript and made the necessary corrections to ensure consistency and accuracy in our use of statistical notation and terminology throughout the document. Specific Corrections Made:

--The "p" on original lines 74 and 78 was italicized (see lines 63 and 68 now).

--We have replaced "p-values" with "p" on the line that was originally 99 (see line 93 now) to provide clarity and adhere to proper terminology.

--The term "xCell" has been correctly capitalized as "XCell" in the sentence that was originally on line 87 (see line 79 now).

Comment 2: When using GO and KEGG to identify the enriched functions and pathways, how do you handle the issue of functional redundancy to prevent over-interpretation caused by a single gene's involvement in multiple pathways?

Response:

Thank you for your insightful comments and for highlighting the need to address the issue of functional redundancy in our analysis using GO and KEGG. We appreciate your concern regarding the potential over-interpretation caused by a single gene's involvement in multiple pathways. To handle redundancy, we set p-value < 0.05 and q-value < 0.4, and defined a 0.7 overlap threshold to merge highly overlapping pathways. We selected the most significant pathway (lowest p-value) as the representative when redundancy was detected. These parameters were applied consistently across both analysis to ensure uniform redundancy handling (see line 67 to 69 now).

Reviewer #2: This manuscript presents a bioinformatics study that identifies potential biomarkers (AURKB, BUB1, CCL2, and TOP2A) and investigates immune cell infiltration patterns in Kawasaki disease (KD). The study offers valuable insights into the molecular mechanisms underlying KD and provides evidence for the potential use of these biomarkers in diagnosis and targeted therapy. However, there are several areas where the manuscript could be improved in terms of structure, clarity, language, and scientific depth. (1)The manuscript lacks a clear, cohesive structure in some sections, particularly in the Results and Discussion. There is frequent overlap, repetition, and mixing of topics;(2) The manuscript contains frequent grammatical errors, awkward phrasing, and informal expressions, which reduce its readability and professionalism. (3)Many ideas, such as the roles of AURKB and BUB1 in mitosis or the enrichment of immune-related GO terms, are repeated across sections without adding new insights. This leads to redundancy. (4) While the manuscript provides detailed descriptions of hub gene functions and pathways, the relevance to KD is not consistently emphasized. Discussions about cancer-related roles of AURKB, BUB1, and TOP2A dominate, while their specific roles in KD are underexplored. Below are partial problems listed in the order they are stated in the paper.

Response:

Thank you for your constructive feedback. We have revised the manuscript as follows: (1)Structure and Coherence: We have reorganized the Results and Discussion sections to ensure clarity and logical flow, removing redundancy and integrating related topics; (2) Language and Grammar: The manuscript has been polished to correct errors and improve readability; (3) Redundancy: We have streamlined repetitive content, focusing discussions on the genes' roles in KD rather than their functions in cancer; (4) Relevance: We have added detailed discussions on the specific roles of AURKB, BUB1, CCL2, and TOP2A in KD, supported by the latest research literature. Below are our responses to the partial problems in the order they are stated.

Comment 1: Use Greek letters consistently (e.g., TNF-α instead of TNF-a) to adhere to academic conventions.

Response:

Thank you for pointing out the need for consistency in the use of Greek letters. We have thoroughly reviewed the manuscript and made the following specific changes: All instances of "TNF-a" have been revised to "TNF-α" to ensure consistency with academic conventions.

Comment 2: "However, patients do no show apparent clinical manifestations, thus delaying diagnosis and treatment."better changes to "However, KD patients often lack apparent clinical manifestations, leading to delayed diagnosis and treatment."

Response:

Thank you for your valuable suggestion regarding the clarity and accuracy of the sentence. We agree with your suggestion and have revised the sentence accordingly. The revised sentence now reads:"However, KD patients often lack apparent clinical manifestations, leading to delayed diagnosis and treatment (see lines 32 to 33 now)."

Comment 3: Some expressions could be made more concise and formal to enhance the academic tone of your writing."The data were cleaned using Limma11packages in R software (4.3.0)." , it could be better revised as "Data normalization and preprocessing were performed using the limma package in R (version 4.3.0)."

Response:

Thank you for your suggestion on improving the conciseness and formality of our writing. We agree with your suggestion and have revised the sentence accordingly. The revised sentence now reads: "Data preprocessing was performed using the limma package in R (version 4.3.0) (see lines 58 to 59 now)"

Comment 4: "log fold change" should be clarified as "log2 fold change" to be consistent with standard terminology.

Response:

Thank you for your suggestion regarding the clarification of terminology. We agree with your suggestion and have clarified "log fold change" as "log2 fold change" throughout the manuscript to ensure consistency with standard terminology (see lines 63, 93 and 94 now).

Comment 5: The current structure is mostly fine, but for better readability and organization, please start each subsection with a brief explanation of the purpose of the analysis. For example:In 2.2, briefly state the purpose of constructing the PPI network (e.g., to identify interactions between differentially expressed genes), etc.

Response:

Thank you for your suggestion on improving the readability and organization of our manuscript. We have revised the manuscript to include a brief explanation of the purpose of each analysis at the beginning of each subsection. For example:

--The purpose of screening DEGs: To identify genes with significant expression changes under different conditions (see lines 60 to 61 now).

--The purpose of GO and KEGG enrichment analysis: To achieve functional interpretation of gene clusters (see lines 66 now).

--The purpose of constructing the PPI network: To identify interactions among DEGs (see lines 71 now).

--The purpose of hub genes: To identify key genes with significant interactions in biological networks (see lines 73 now).

--The purpose of Cell Infiltration analysis: The XCell algorithm was then used to evaluate the infiltration scores of 29 immune and non-immune cells (see lines 79 to 80 now).

--The purpose of Correlation analysis and PCA: To assess the correlation between hub genes and immune cells, and to assess the efficacy of a node-gene signature in distinguishing KDs from NCs (see lines 84 to 85 now).

Comment 6: Some expressions are too general or lack specificity. Consider the following refinements:

2.2: PPI Network Construction: Specify the STRING database threshold (e.g., combined score cutoff for significance); Explain how hub genes were identified (e.g., via centrality metrics like degree or betweenness). Include the version of Cytoscape for reproducibility.

Response:

Thank you for your valuable feedback regarding the specificity of our expressions in Section 2.2. We have made the following refinements to address your concerns:

--STRING Database Threshold: We have specified the STRING database threshold as ≥0.7 in the revised text (see lines 68 now);

--Hub Gene Identification: We have clarified that hub genes were identified as those with degree centrality above the 90th percentile in the PPI network, using the igraph R package (see lines 73 to 75 now);

--Cytoscape Version: We thank the reviewer for raising this point. Upon further review, we realized that Cytoscape was mentioned in the manuscript but was not actually used in our study. To ensure accuracy and reproducibility, we have removed all references to Cytoscape and clarified the tools we actually employed. We apologize for any confusion this may have caused.

2.3: Immune Cell Infiltration Analysis: Shorten and simplify the description of xCell without losing its technical essence. State how input data were normalized or preprocessed before applying xCell. Reorganize for better readability: explain the analysis, significance threshold, and visualization in sequence.

Response:

Thank you for your suggestions regarding Section 2.3 on Immune Cell Infiltration Analysis. We have revised the section to address your concerns as follows:

--Shortened and Simplified Description of xCell: We have shortened and simplified the description of the XCell algorithm while retaining its technical essence. The revised sentence now reads: "We then used the XCell algorithm to evaluate infiltration scores for 29 immune and non-immune cells" (see lines 79 to 80 now).

--Normalization and Preprocessing of Input Data: To address the reviewer's concerns, we have normalized and preprocessed the GSE73461 dataset by removing genes with zero average expression, applying scaling normalization, and excluding IL4 due to low post-normalization expression (see lines 77 to 79 now).

--Reorganized for Better Readability: We have reorganized the text to improve readability by sequentially explaining the analysis, significance threshold, and visualization. The revised text now clearly outlines the steps: data preparation and normalization, application of the XCell algorithm to evaluate infiltration scores, identification of significant differences at P < 0.05, and visualization using heatmaps and violin plots (see lines 77 to 82 now).

2.4: Correlation Analysis and PCA: Specify "hub genes" and "immune cells" explicitly in the correlation analysis description. Use consistent package names (e.g., "ggplot2" in backticks). For Spearman analysis: indicate how correlation coefficients were interpreted (e.g., thresholds for "strong" or "weak" correlations).

Response:

Thank you for your detailed feedback regarding Section 2.4. We have made the following revisions to address your concerns:

--Explicit Specification of "Hub Genes" and "Immune Cells": We have revised the description to explicitly specify “hub genes” and “immune cells” in the correlation analysis. The updated text now reads: “To evaluate the correlation between candidate hub genes and infiltrating immune cells, and to assess the efficacy of a node-gene signature in distinguishing KDs from NCs, we used the Spearman correlation method (P < 0.05)” (see lines 84 to 86 now). We acknowledge that the XCell algorithm includes both immune and non-immune cells, but our analysis specifically focused on immune cells in this context.

--Consistent Package Names: We have standardized the formatting of package names throughout the manuscript, using backticks for consistency (e.g., ggplot2) (see lines 87 and 90 now). Thank you for pointing this out.

--Interpretation of Correlation Coefficients: We have added a description of how correlation coefficients were interpreted in the Spearman analysis. The thresholds are now clearly defined as follows: strong (|r| ≥ 0.7), moderate (0.4 ≤ |r| < 0.7), and weak (0.1 ≤ |r| < 0.4) (see lines 86 to 87 now).

Comment 7: The logical flow of the results section is unclear, and the connections between subsections are weak and the results feel fragmented, and there is no clear summary of key findings in some sections. Please reorganize the results section to create a more coherent flow. Use a logical order such as:

--Identification of DEGs: Provide the results of DEG analysis and briefly explain its significance.

--GO and KEGG Enrichment Analysis: Highlight the biological relevance of the enriched pathways and terms.

--PPI Network Construction and Hub Gene Identification: Emphasize the key findings related to hub genes.

--Immune Cell Infiltration Analysis: Explain the immune landscape differences in KD.

--Correlation Analysis and PCA: Highlight the relationships between immune cells and hub genes, and show how PCA distinguishes KD and NC samples.

Response:

Thank you for your valuable feedback regarding the logical flow of the results section. We have thoroughly reorganized the Results section to improve logical flow and coherence. We have structured the subsections in a more logical order and added transitions to strengthen the connections between them. Additionally, we have included clear summaries of key findings in each section to provide a better overview of the results. Thank you for your valuable suggestions. The specific changes are as follows:

--Identification of DEGs: We have revised the section on DEG identification to provide more detailed results and explain their significance. The updated text now reads: “In this study, we identified 1229 DEGs with P-values < 0.05 and |log2 FC|≥ 2, including 720 up-regulated and 509 down-regulated genes. Notably, those with |log2 FC| > 8 are particularly important for understanding specific biological processes or disease states and require further study (Fig. 1)” (see lines 93 to 95 now).

--GO and KEGG Enrichment Analysis: We have revised the section on GO and KEGG enrichment analysis to better highlight the biological relevance of the enriched pathways and terms. The updated text now emphasizes how these findings relate to Kawasaki disease (KD) and their significance in understanding its molecular mechanisms. Specifically, we discuss the links to acute inflammatory response and plasma membrane components in GO analysis, as well as the enrichment in cytokine interactions and PI3K-Akt signaling in KEGG analysis. These results support KD's known inflammatory and immune-mediated nature and provide insights into potential therapeutic targets (see lines 97 to 104 now).

--PPI Network Construction and Hub Gene Identification: We have revised the section on PPI network construction and hub gene identification to emphasize the key findings related to hub genes. The updated text now clearly highlights the identification of five prominent candidate hub genes (AURKB, BUB1, CCL2, IL4, and TOP2A) and their significant expression levels, underscoring their pivotal role in KD pathogenesis (see lines 107 to 110 now).

--Immune Cell Infiltration Analysis: We have revised the section on im

---

## [Decision Letter · Decision Letter 1]

24 Apr 2025

Identification of Potential Biomarkers associated with Immune Cell Infiltration Patterns in Kawasaki Disease via Bioinformatics

PONE-D-24-55722R1

Dear Dr. Wang,

We’re pleased to inform you that your manuscript has been judged scientifically suitable for publication and will be formally accepted for publication once it meets all outstanding technical requirements.

Kind regards,

Xiaosheng Tan

Academic Editor

PLOS ONE

Additional Editor Comments (optional):

Reviewers' comments:

Reviewer's Responses to Questions

**Comments to the Author**

1. If the authors have adequately addressed your comments raised in a previous round of review and you feel that this manuscript is now acceptable for publication, you may indicate that here to bypass the “Comments to the Author” section, enter your conflict of interest statement in the “Confidential to Editor” section, and submit your "Accept" recommendation.

Reviewer #1: All comments have been addressed

Reviewer #2: All comments have been addressed

Reviewer #3: All comments have been addressed

2. Is the manuscript technically sound, and do the data support the conclusions?

Reviewer #1: (No Response)

Reviewer #2: Yes

Reviewer #3: (No Response)

3. Has the statistical analysis been performed appropriately and rigorously? 

Reviewer #1: (No Response)

Reviewer #2: Yes

Reviewer #3: (No Response)

4. Have the authors made all data underlying the findings in their manuscript fully available?

Reviewer #1: (No Response)

Reviewer #2: Yes

Reviewer #3: (No Response)

5. Is the manuscript presented in an intelligible fashion and written in standard English?

Reviewer #1: (No Response)

Reviewer #2: Yes

Reviewer #3: (No Response)

6. Review Comments to the Author

Reviewer #1: (No Response)

Reviewer #2: (No Response)

Reviewer #3: (No Response)

7. PLOS authors have the option to publish the peer review history of their article (what does this mean? ). If published, this will include your full peer review and any attached files.

**Do you want your identity to be public for this peer review?** For information about this choice, including consent withdrawal, please see our Privacy Policy .

Reviewer #1: No

Reviewer #2: No

Reviewer #3: No

---

## [Editor Report · Acceptance letter]

PONE-D-24-55722R1

PLOS ONE

Dear Dr. Wang,

I'm pleased to inform you that your manuscript has been deemed suitable for publication in PLOS ONE. Congratulations! Your manuscript is now being handed over to our production team.

Kind regards,

on behalf of

Dr. Xiaosheng Tan

Academic Editor

PLOS ONE